# Ryanodine receptor 2 inhibition reduces dispersion of cardiac repolarization, improves contractile function, and prevents sudden arrhythmic death in failing hearts

Pooja Joshi[1], Shanea Estes[1], Deeptankar DeMazumder[2,3,4,5], Bjorn C Knollmann[1], Swati Dey[1]*

[1]Division of Clinical Pharmacology, Department of Medicine, Vanderbilt University Medical Center, Nashville, United States; [2]Section of Cardiac Electrophysiology, Division of Cardiology, Department of Internal Medicine, Veterans Affairs Pittsburgh Health System, Pittsburgh, United States; [3]Section of Cardiac Electrophysiology, Division of Cardiology, Department of Internal Medicine, University of Pittsburgh School of Medicine and University of Pittsburgh Medical Center, Pittsburgh, United States; [4]Department of Surgery, University of Pittsburgh School of Medicine, Pittsburgh, United States; [5]McGowan Institute for Regenerative Medicine, University of Pittsburgh School of Medicine, Pittsburgh, United States

*For correspondence:
swati.dey@vumc.org

**Competing interest:** The authors declare that no competing interests exist.

**Abstract** Sudden cardiac death (SCD) from ventricular tachycardia/fibrillation (VT/VF) is a leading cause of death, but current therapies are limited. Despite extensive research on drugs targeting sarcolemmal ion channels, none have proven sufficiently effective for preventing SCD. Sarcoplasmic ryanodine receptor 2 (RyR2) $Ca^{2+}$ release channels, the downstream effectors of sarcolemmal ion channels, are underexplored in this context. Recent evidence implicates reactive oxygen species (ROS)-mediated oxidation and hyperactivity of RyR2s in the pathophysiology of SCD. We tested the hypothesis that RyR2 inhibition of failing arrhythmogenic hearts reduces sarcoplasmic $Ca^{2+}$ leak and repolarization lability, mitigates VT/VF/SCD and improves contractile function. We used a guinea pig model that replicates key clinical aspects of human nonischemic HF, such as a prolonged QT interval, a high prevalence of spontaneous arrhythmic SCD, and profound $Ca^{2+}$ leak via a hyperactive RyR2. HF animals were randomized to receive dantrolene (DS) or placebo in early or chronic HF. We assessed the incidence of VT/VF and SCD (primary outcome), ECG heart rate and QT variability, echocardiographic left ventricular (LV) structure and function, immunohistochemical LV fibrosis, and sarcoplasmic RyR2 oxidation. DS treatment prevented VT/VF and SCD by decreasing dispersion of repolarization and ventricular arrhythmias. Compared to placebo, DS lowered resting heart rate, preserved chronotropic competency during transient β-adrenergic challenge, and improved heart rate variability and cardiac function. Inhibition of RyR2 hyperactivity with dantrolene mitigates the vicious cycle of sarcoplasmic $Ca^{2+}$ leak-induced increases in diastolic $Ca^{2+}$ and ROS-mediated RyR2 oxidation, thereby reducing repolarization lability and protecting against VT/VF/SCD. Moreover, the consequent increase in sarcoplasmic $Ca^{2+}$ load improves contractile function. These potentially life-saving effects of RyR2 inhibition warrant further investigation, such as clinical studies of repurposing dantrolene as a potential new therapy for heart failure and/or SCD.

### eLife assessment

This **important** study examined the use of dantrolene, a Ryanodine Receptor stabilizer, in slowing pathological progression of pressure-overload heart failure in a guinea pig model and reducing arrhythmias. **Convincing** data were collected and analyzed using validated methodology and can be used as a starting point for future studies of dantrolene in Ca2+ handling in ROS production and further deterioration of cardiac function in chronic heart failure.

## Introduction

SCD in heart failure (HF) is the leading cause of death in the modern world. Current therapies for preventing SCD due to VT/VF are limited. For example, ICDs, the only effective therapy for VT/VF and SCD, are palliative, expensive, and pose several risks. Moreover, many patients suffer from VT/VF and SCD in the early stages of HF before they become eligible to receive an ICD (*Aziz et al., 2021*; *Butrous et al., 1992*; *Krishnan et al., 2018*; *Richardson et al., 2018*; *Vaseghi et al., 2017*; *Yalin et al., 2021*; *Zhou et al., 2019*; *Zanoni et al., 2017*; *Tygesen et al., 1999*). Antiarrhythmic drugs, which target ion channels, may confer acute benefits but can also be proarrhythmic, triggering VT/VF. Importantly, most antiarrhythmic drugs worsen survival over the longer term (*Echt et al., 1991*; *The Cardiac Arrhythmia Suppression Trial II Investigators, 1992*). Thus, there is a pressing need to identify new antiarrhythmic targets and develop new therapies for SCD.

A growing body of evidence indicates that sympathetic stress-induced hyperactivity of the RyR2, located in the SR of cardiac myocytes, causes SR $Ca^{2+}$ leak, which contributes to the pathogenesis of VT/VF and SCD (*Fischer et al., 2013*; *Alvarado and Valdivia, 2020*; *Wehrens et al., 2006*; *Eisner et al., 2009*). During systole, the release of $Ca^{2+}$ by the RyR2 plays a pivotal role in myocyte excitation-contraction coupling. The degree of contractility is determined by the amount of $Ca^{2+}$ in the SR and by the timing and magnitude of $Ca^{2+}$ release. In diseased and/or stressed hearts, PKA-mediated hyper-phosphorylation and ROS-mediated oxidation results in a leaky RyR2 (*Echt et al., 1991*), which allows $Ca^{2+}$ to leak out of SR during diastole. The consequent reduction in SR $Ca^{2+}$ load weakens systolic contraction. The subsequent rise in cytosolic $Ca^{2+}$ levels also leads to DADs, which can trigger atrial and ventricular tachyarrhythmias (*Al-Khatib et al., 2018*; *Wehrens, 2007*; *Hamilton et al., 2020*; *Liu et al., 2020*). Recent evidence suggests that dantrolene, a RyR2 inhibitor, reduces the inducibility of atrial fibrillation in animals with myocardial infarction (*Azam et al., 2021*; *Zamiri et al., 2014*; *Chou et al., 2014*; *Meissner et al., 1999*). The effect of RyR2 inhibition on ventricular arrhythmias and non-ischemic HF has been unexplored.

We hypothesized that RyR2 inhibition of failing arrhythmogenic hearts reduces sarcoplasmic $Ca^{2+}$ leak and repolarization lability, mitigates VT/VF/SCD and improves contractile function. The present study employs a guinea-pig model that recapitulates key features of human non-ischemic HF including spontaneous episodes of VT/VF resulting in SCD (*Dey et al., 2018*; *Liu et al., 2014*; *Liu et al., 2021*). We examine the effects of chronic dantrolene treatment on the incidence of VT/VF and SCD incidence as non-ischemic HF develops and progresses. In cross-over studies, we determine whether RyR2 inhibition reverses HF and SCD risk. The findings have important implications for repurposing dantrolene, a clinically-used RyR2 inhibitor, as a potential new therapy for HF and SCD.

## Results

### Inhibition of RyR2 leak with dantrolene therapy prevents VT/VF and SCD

To study the effects of RyR2 leak inhibition on SCD and HF, we leveraged a pressure overload guinea pig model of HF/SCD with spontaneous arrhythmias. Unique to this model is the high incidence of premature ventricular contractions (PVCs) and VT/VF, leading to SCD in about two-thirds of the animals within 4 weeks. Notably, the SCD occurs early on, in the first 2 weeks over the course of cardiac hypertrophy, even before the heart dilates or progresses into failure (*Dey et al., 2018*; *Liu et al., 2014*). Hence, the primary endpoint of our study was SCD, and the secondary endpoint was cardiac contractility measured by echocardiography at four weeks. To determine if Dantrolene sodium (DS) treatment can be used as a potential therapeutic agent, we treated the HF/SCD animals with

**eLife digest** Each year, more than 300,000 people experience cardiac arrest or sudden cardiac death. Sudden cardiac death is caused by irregular heartbeats known as ventricular tachycardia or ventricular fibrillation, which prevent the heart from pumping blood.

During a regular heart rhythm, the heart muscles contract and relax, regulated by a coordinated rise and fall of calcium ions within heart cells. In the cells of diseased hearts, on the other hand, calcium leaks out of a compartment known as the sarcoplasmic reticulum in an uncontrolled manner. This happens because an ion channel in the membrane of the sarcoplasmic reticulum known as ryanodine receptor 2 becomes hyperactive and releases calcium in an uncontrolled manner. This abnormal calcium release leads to irregular calcium waves, which can make the heart's electrical properties unstable, causing ventricular tachycardia, ventricular fibrillation and sudden cardiac death.

Joshi et al. tested whether dantrolene, a molecule that blocks ryanodine receptor 2, can stop calcium leaks from the sarcoplasmic reticulum and prevent lethal arrhythmias and sudden cardiac death in failing hearts. To investigate this, Joshi et al. induced heart failure in guinea pigs that have abnormal heart calcium signalling similar to human heart failure, and then treated the animals with either dantrolene or a placebo.

The results indicate that blocking ryanodine receptor 2 hyperactivity with dantrolene prevents lethal arrhythmias and sudden cardiac death by blocking calcium leaks and by preventing the instability of the electrical properties of the heart. Additionally, Joshi et al. found that dantrolene also improved the diseased heart's ability to pump adequate amounts of blood, allowing failing hearts to meet increased cardiovascular demands, and thereby improving the heart's overall function.

The proposed studies come from a strong clinical need to improve bad outcomes in people who keep having fatal heart rhythm episodes despite getting the best medical care. Many heart failure patients are plagued by recurrent defibrillator shocks to abort sudden cardiac death from relentless lethal heart rhythms. These shocks are painful, injure the heart, and worsen the quality of life. Unfortunately, management options are extremely limited for these patients.

The findings of Joshi et al. indicate that dantrolene may be a potential treatment for people with fatal heart rhythms who are at risk of sudden cardiac death and could have a positive impact on these people's quality of life. However, before this can happen, dantrolene will first have to be thoroughly tested to ensure effectivity and safety in humans. In any case, Joshi et al. have opened a new avenue in the search for medications to treat deadly arrhythmias and sudden cardiac death.

and without oral DS in the early stages of heart failure (*Figure 1A*). HF animals exhibit frequent PVCs that often trigger sustained and non-sustained VT and VF, causing SCD (*Figure 1B*). DS treatment not only drastically reduced the rate of PVCs in early and late stages of HF (*Figure 1C*), but also improved survival (*Figure 1D*). Compared to the HF group, the HF + DS group had a significantly higher rate of survival (55% HF vs 80% in HF +DS, p<0.05).

## Dantrolene therapy improves cardiac function and prevents HF

Cardiac structure and function were preserved in HF animal treated with DS as compared to the HF group. In contrast to the HF group, the HF+DS group did not display a significant dilation of the LV at the 4 week endpoint (*Figure 2A*). Echocardiography measures (*Figure 2B*) show that the relative wall thickness (RWT), which is a measure of the LV dimensions and is a marker for adverse events in LV dysfunction, was significantly lower in HF group as compared to control (*Figure 2C*, p<0.05). Progressive decline of the RWT causes the end systolic volume (ESV) and the end diastolic volume (EDV) to increase and cardiac output to decrease. Reduced contractility and increased afterload both reduce the ejection fraction and increase end-systolic volumes in failing hearts (*Figure 2D–F*). Additionally, EDV increased (not shown) without an increase in the stroke volume. To summarize, chronic DS therapy in HF prevented the thinning of the LV walls and the eventual decline in cardiac contractility.

## Dantrolene prevents myocardial structural and molecular damage

Analysis of the myocardial structure showed an increase in interstitial fibrosis in the HF group compared to the normal controls. Whereas the fibrotic tissue was twofold higher in the untreated

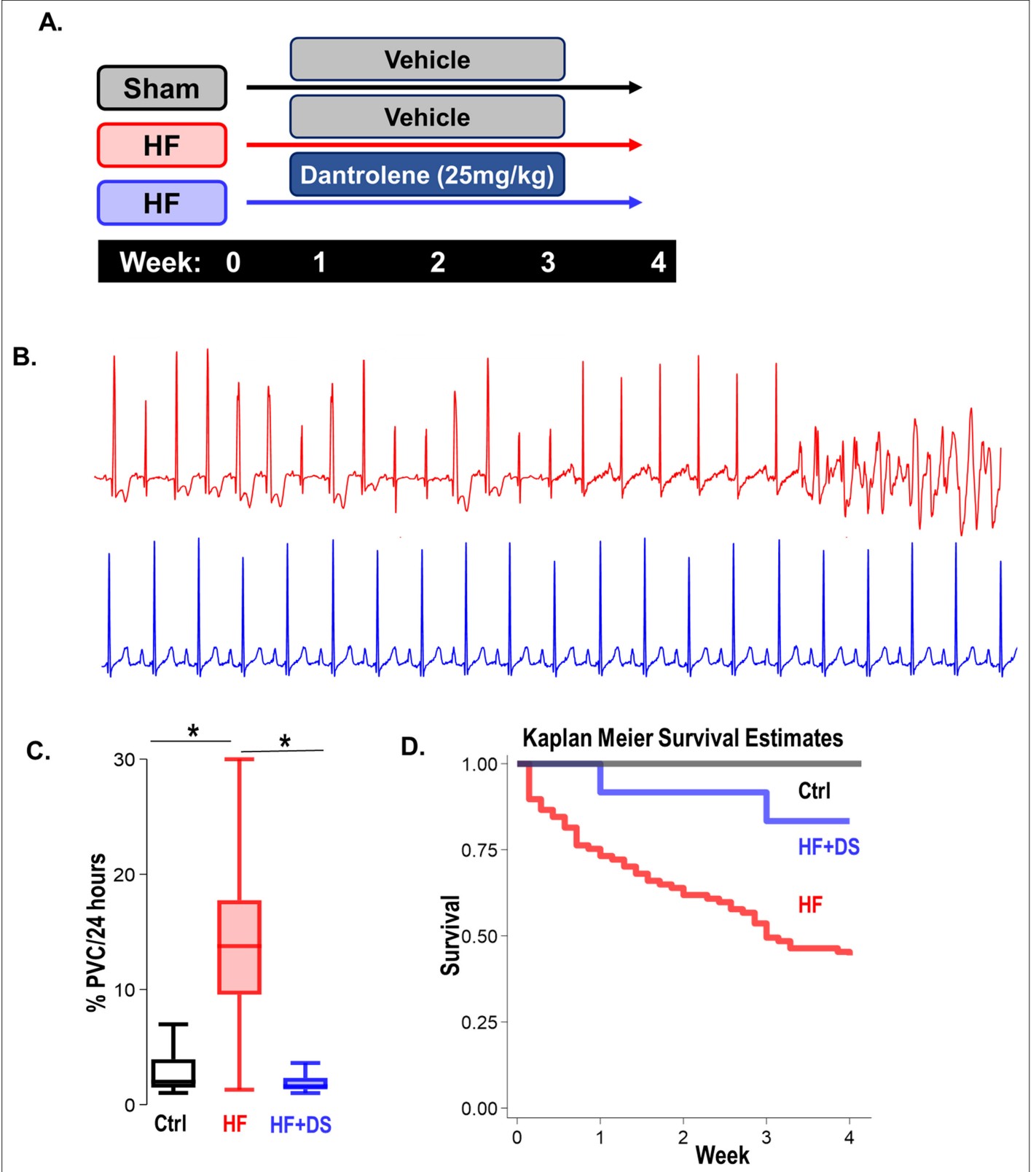

**Figure 1.** Dantrolene mitigates ventricular tachycardia/fibrillation (VT/VF) and prevents sudden cardiac death (SCD) in vivo. (**A**) Schematic of experimental design. (**B**) Representative Electrocardiogram (ECG) tracings from 24 hr telemetry recordings. Spontaneous incidences of premature ventricular contractions (PVCs), VT, and VF were suppressed in heart failure (HF) animals with chronic dantrolene (DS) treatment. (**C**) DS therapy lowered PVC burden (p<0.001) in HF animals (N: Control, black = 15, HF, red = 20, HF+DS, blue = 8). (**D**) Kaplan Meier's survival plot shows that over 50% of

*Figure 1 continued on next page*

*Figure 1 continued*

the animals in the vehicle group (HF, N=68, red) group experienced SCD. Chronic DS treatment (HF+DS, N=10, blue; Ctrl, N=10, black) prevented VT and VF in heart failure models and mitigated SCD in 80% of the heart failure animals (p<0.05). A p<0.05 was considered significant; two-tailed log-rank analysis was performed on each group for each measure.

HF group compared to normal hearts, the HF+DS group displayed lower fibrotic tissue thereby confirming that DS therapy was delaying the progression of HF, keeping the myocardial architecture intact (*Figure 3A–B*). Interstitial fibrosis is a histological hallmark associated with the progression of HF and can cause lethal arrhythmias due to reentry circuits.

In our previous report, we noted HF-associated changes in both expression and phospho-proteome of key ion channels, transporters, and excitation contraction-coupling proteins. In particular, hyper-phosphorylation of several RyR2 sites was noted in failing hearts when compared to normal (*Dey et al., 2018*; *Figure 3—figure supplement 1*). A targeted antioxidant therapy prevented these modifications, pointing to oxidative damage as the primary cause. Therefore, we tested if DS therapy prevented the downstream oxidative damages by quantifying the oxidation profile of RyR2 with and without DS. RyR2 oxidation was significantly reduced in HF+DS group as compared to HF group (*Figure 3C*).

## Dantrolene treatment decreases heart rate and improves chronotropic competency in HF/SCD

As per our experimental protocol, all groups of animals received a daily low dosage of isoproterenol bolus for 1 hr. As anticipated, the heart rate was elevated in response to β-adrenergic stimulation and recovered to resting heart rate 3–4 hr later. We assessed ECG parameters at the resting heart rate and

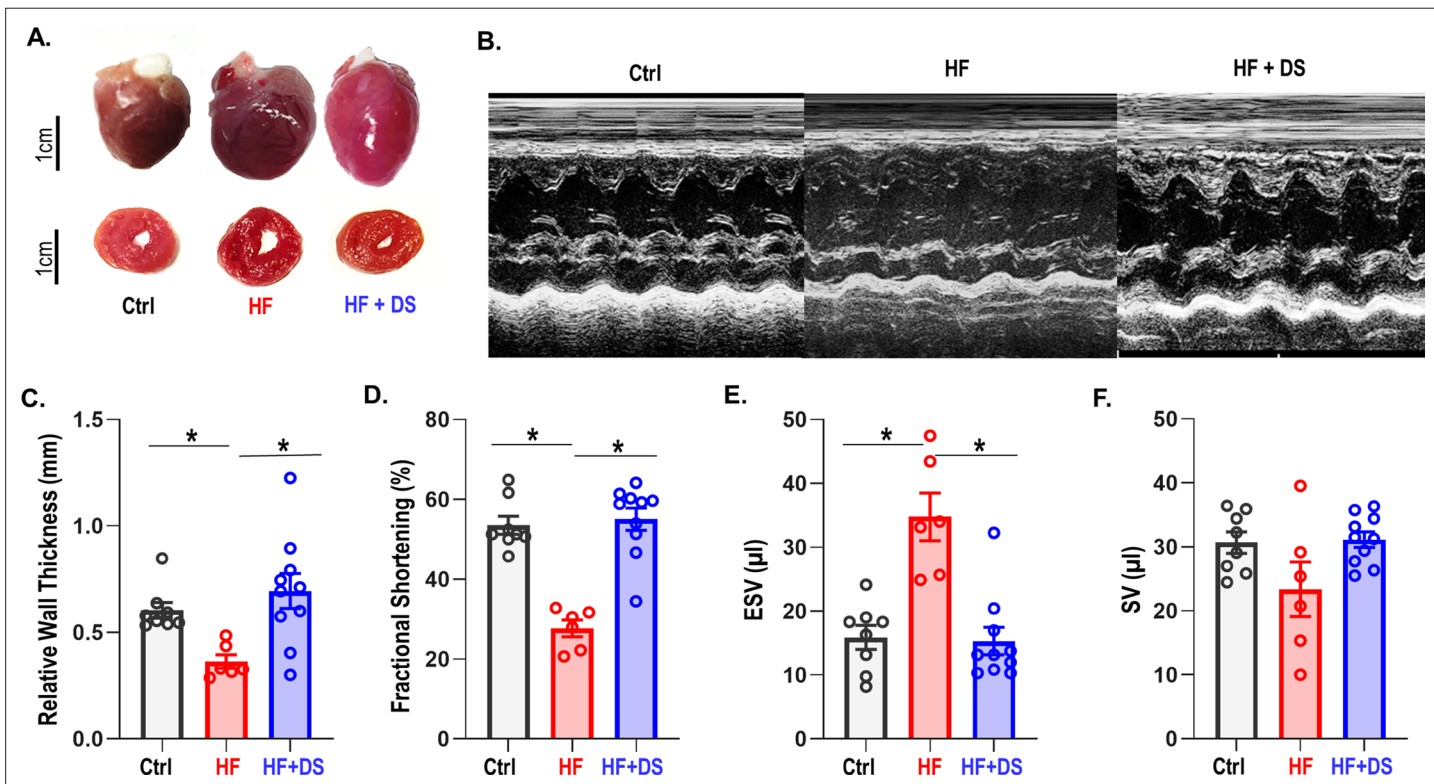

**Figure 2.** Dantrolene improves cardiac function and reverses heart failure. (**A**) The hearts in the vehicle groups are enlarged, as shown in representative photos of gross hearts (top) and matching cross-sections (bottom) after 4 weeks. In the heart failure (HF) animals, the left ventricular (LV) cavity is bigger, and the LV free wall is thinned. Treatment with dantrolene (DS) stops the LV walls from thinning. Chronic DS group hearts were normal in size. (**B**) Representative M-mode echocardiography images from all groups (**C–F**) Echocardiography parameters at 4 weeks. HF animals showed significant (p<0.005, N: Control = 15, HF = 20, HF+DS = 8) loss of cardiac function with a decline in fractional shortening and relative wall thickness and an increase in stroke volume (SV) and end systolic volume (ESV). DS therapy prevented pressure overload dependent structural remodeling of the heart (p<0.005).

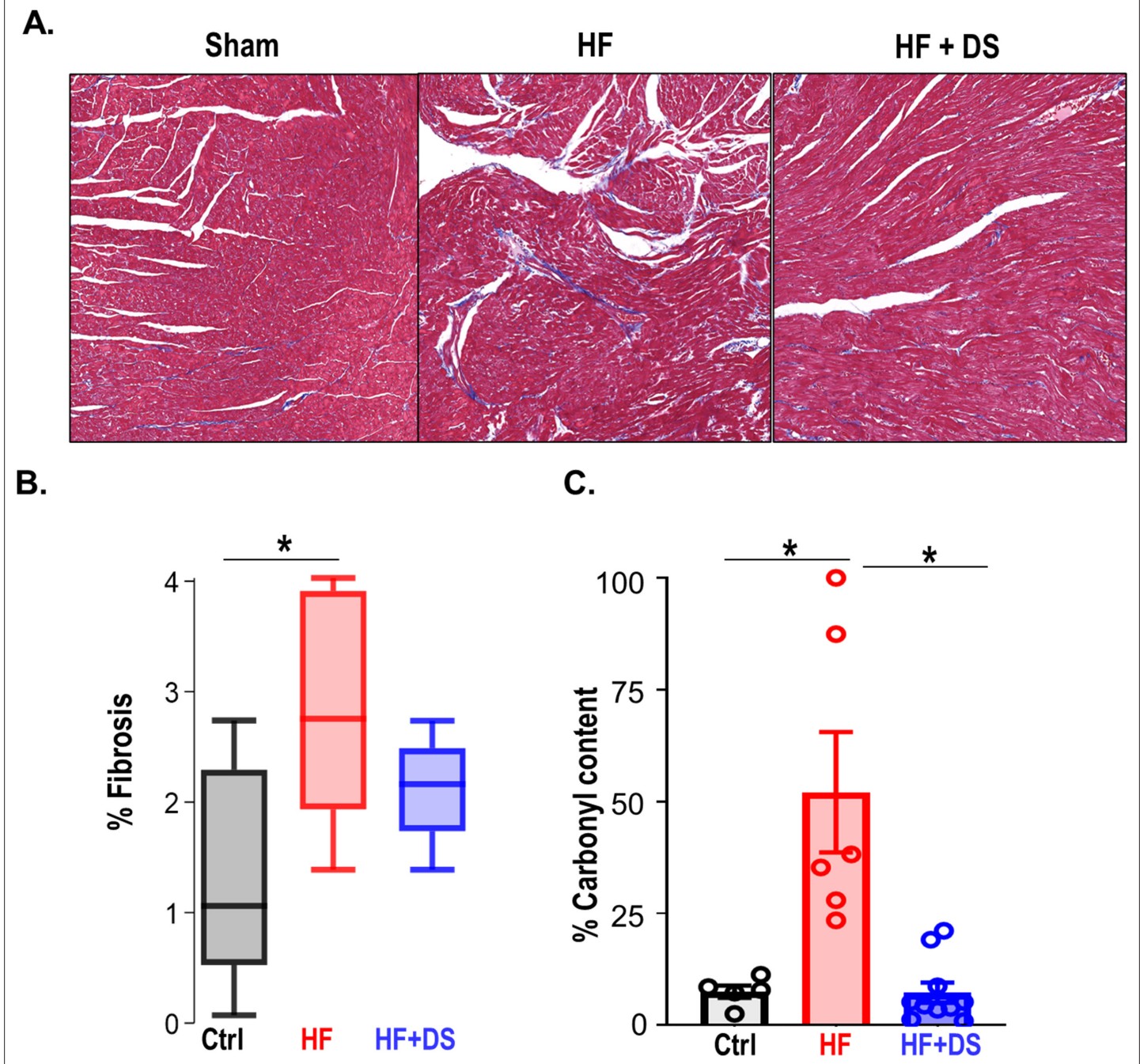

**Figure 3.** Dantrolene reverses tissue damage and oxidation profile of ryanodine receptor 2 (RyR2). (**A**) Representative image of Masson trichrome staining (318 x 344 pixels) in Ctrl, heart failure (HF) & HF+DS hearts (N=3) showing high fibrotic regions as (**B**) summarized. (**C**) The percent carbonyl content/mg of protein was lower in dantrolene (DS)-treated heart failure animals, showing reversal of RyR2 oxidation profile with DS therapy in HF animals (p<0.005, N≥4). A p<0.05 was considered significant; two-tailed log-rank analysis was performed on each group for each measure.

The online version of this article includes the following figure supplement(s) for figure 3:

**Figure supplement 1.** The Guinea Pig model of heart failure (HF)/(sudden cardiac death) SCD shows a remodeled and hyperactive ryanodine receptor 2 (RyR2).

during post β-AR stress recovery (*Figure 4A*). As expected, to adequately respond to the metabolic demand and increased workload, failing hearts fine-tune the resting heart rate. The HF animals had a significant elevation of the resting heart rate (269 ± 10 bpm) as compared to normal controls (251 ± 5, p<0.005). HF+DS animals, however, demonstrated a significantly lower heart rate compared to

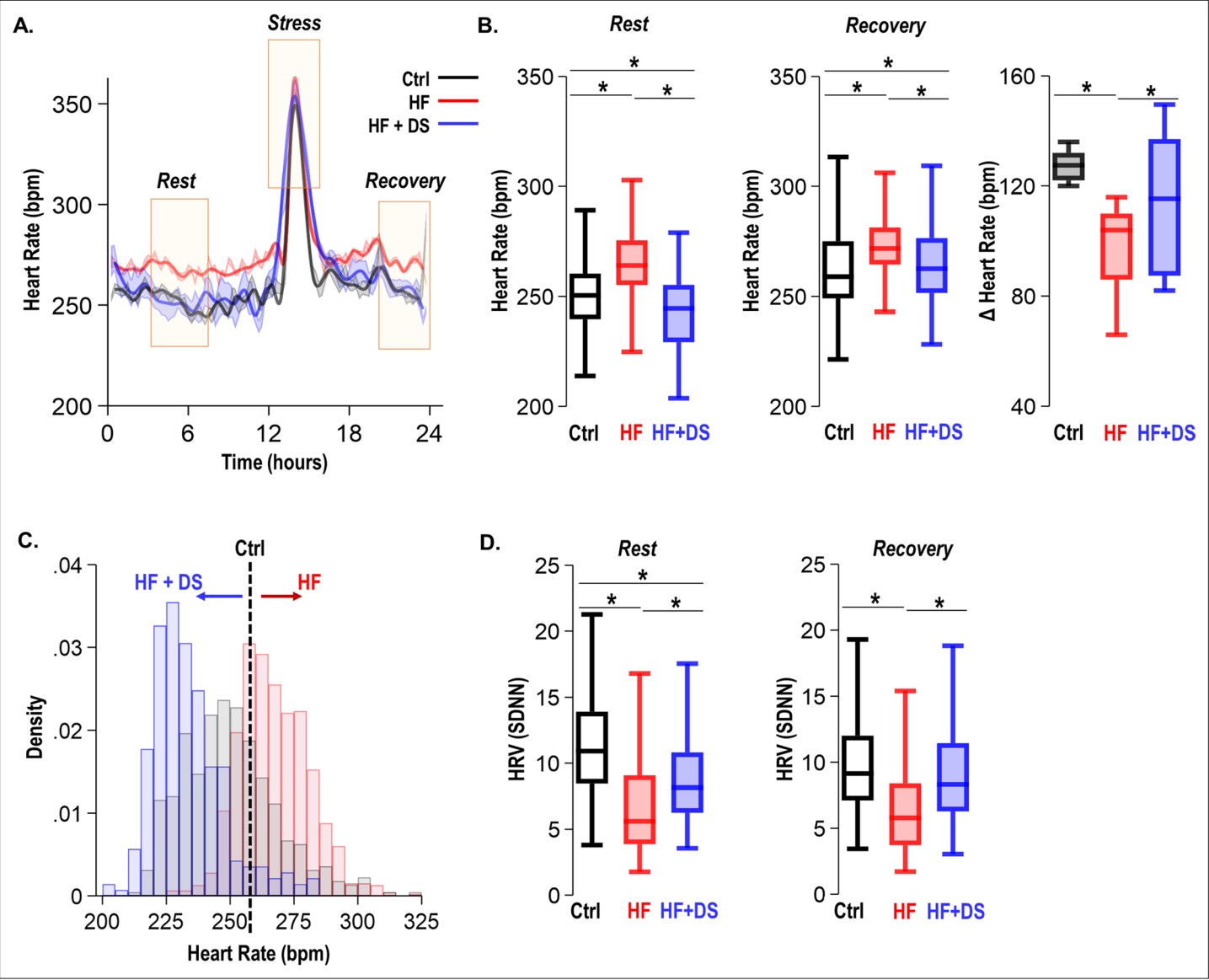

**Figure 4.** Dantrolene treatment decreases heart rate and improves chronotropic competency in heart failure (HF). (**A**) Plot shows heart rate derived from 24 hr continuous electrocardiogram (ECG) recordings. The animals were subjected to mild transient β-AR stress for 1 hr. Continuous ECG analysis was performed at the following time points: resting heart rate (pre-stress); transient stress and, post-stress recovery (4 hr post-stress) (**B**) The box plots summarize HR during the resting and recovery phases. The dantrolene (DS) treated group (HF+DS) show a lower heart rate (HR) both at rest and post-stress recovery. The Δ Heart Rate (bpm) indicates the net increase in HR from resting HR to peak HR in response to transient β-AR stress. Failing hearts displayed higher resting HR and blunted chronotropic response to stress (p<0.02, N≥7 for all models). (**C**) Histogram shows resting HR distribution for all groups. Resting heart rate is decreased in HF+DS animals. However, the peak heart rates (during stress) are higher in HF+DS compared to heart failure (HF) (N; Ctrl = 8, HF = 10; HF+DS = 7). (**D**) DS increases heart rate variability (HRV) both at rest and during stress (p<0.05, N≥7 for all models). A p<0.05 was considered significant; two-tailed log-rank analysis was performed on each group for each measure.

the HF group, which was similar to that of normal controls (243 ± 5 bpm, p<0.005) (*Figure 4B*). Paired analysis of the heart rate for each subject over 4 weeks shows that HF animals experienced an increase in resting heart rate by 20–40 bpm. Whereas HF+DS animals experienced a decrease in heart rate by a minimum of 20–30 bpm. The histogram in *Figure 4C* depicts the distribution of heart rate over a 24 hr period from the normal, HF, and HF+DS groups, demonstrating the decreased heart rate in the HF+DS group.

Despite increased resting heart rate, the peak response to transient β-AR stress was much lower in HF than HF+DS group, indicative of the therapeutic role of DS in restoring chronotropic competence in failing hearts. Chronotropic competence describes the capacity of the heart to increase its rate in

response to increased demand effectively. The average increase in heart rate during transient stress in HF group (96 ± 5 bpm) was significantly lower (p<0.05) than HF+DS group (114 ± 9 bpm). Furthermore, the heart rate variability (HRV, SDNN) was higher in HF+DS group (9.00 ± 0.21) at rest and post-stress recovery (*Figure 4D*), similar to normal animals as compared to HF (7.28 ± 0.16; p<0.005). DS treatment not only restored the chronotropic response to stress but also reduced the incidences of VT/VF and SCD. To understand this protective mechanism, we next tested the hypothesis that DS prevents VT/VF and SCD by preventing repolarization abnormalities.

## Dantrolene prevents repolarization abnormalities, normalizes the QT variability, and reduces T-wave heterogeneity

Detailed analysis of the ECG features revealed how inhibition of the RyR2 leak with DS prevents VT/VF and SCD. QT variability and prolongation are known to have prognostic significance in failing hearts. While prolonged QTc is considered a risk factor for arrhythmias, QT heterogeneity as an add-on is a surrogate for lethal VT/VF (*Cvijić et al., 2018*). The HF group displayed prolonged QTc, increased QT variability and repolarization abnormalities (*Figure 5A*). The guinea pig model of HF exhibits prolonged QTc, as shown in our previous studies (*Dey et al., 2018*; *Liu et al., 2014*). The QTc of HF+DS group was significantly shorter than HF group and was similar to normal animals at resting heart rate (HF vs HF+DS; 175.59 ± 4 vs 155.07 ± 10 msec, p<0.05). DS treatment not only prevented QTc prolongation (*Figure 5B*), but also significantly reduced QT variability. *Figure 5A* also illustrates the temporal dispersion of repolarization by superimposing T waves from a 24-hr period of ECG recording at week four. QT dispersion was reduced in the DS-treated group, thereby imparting greater electrical stability to the heart. The QT variability at resting heart rate and post-stress recovery are lower in DS-treated HF animals (*Figure 5C-D*) (HF vs HF+DS; -0.35 ± 0.02 vs -0.58 ± 0.02, p<0.001). Inhibiting RyR2 with DS reduced the dispersion of repolarization/QT variability, demonstrating a central role of diastolic calcium in increasing cardiac repolarization lability, and terminating triggered activity.

## Crossover studies. Dantrolene after HF development mitigates VT/VF and prevents SCD

A crossover study was designed where (i) only vehicle was administered until the endpoint i.e., week 4 (HF) (ii) DS was administered at the onset of aortic constriction until chronic contractile dysfunction is evident in the HF group (usually 3 weeks post-banding; HF+/−DS) or (iii) after HF had already developed (3 weeks post-banding; HF−/+DS group) (*Figure 6A*). HF animals were randomized and were either treated with DS after chronic HF had developed or at the time of aortic banding. In the HF+/−DS group, none of the animals experienced SCD, including the period when DS therapy was stopped (*Figure 6B*). Serial echocardiography shows that the FS% in the HF group declines over a period of 4 weeks. DS prevented a decline in cardiac function. Interestingly, HF+/−DS group, the fractional shortening does not decline significantly after stopping DS therapy. This could possibly be an effect of the washout period (*Figure 6C*) (FS; pre vs post: 43.35 ± 0.45 vs 53.10 ± 3.16, p: ns). In contrast, the HF−/+DS groups experienced SCD similar to that of HF before DS therapy was started. However, after crossover to DS treatment, no further SCD occurred. Notably, HF−/+DS group, which showed poor cardiac function at week 3 was noted to quickly recover and normalize FS% (*Figure 6*, FS%; pre vs post: 32.43 ± 3.48 vs 43.10 ± 0.87).

We next examined the ECG parameters (*Figure 6E-F*) in HF−/+DS group at week 1 (post-banding, initial stages of HF), Week 3 (chronic HF), and Week 4 (after starting DS therapy). PVC burden significantly increases in the HF−/+DS group pre-DS therapy but declines sharply post-therapy. We observed the opposite effect in the HF+/−DS group, where PVC burden increases after DS therapy is stopped (*Figure 6—figure supplement 1*). The resting heart rate during chronic HF is significantly higher than pre-HF. After starting DS therapy in chronic failing hearts, we observed a decrease in resting HR to pre-HF levels. Similar trends were seen in the post-stress recovery period. Similarly, HRV improved significantly after the start of DS. The QTc interval was shortened after DS treatment as compared to during pre or chronic HF. The QT variability was also reduced after starting DS. Taken together, RyR2 inhibition with DS is effective both in the early and late stages of HF, preventing VT/VF and SCD and improving cardiac contractile function.

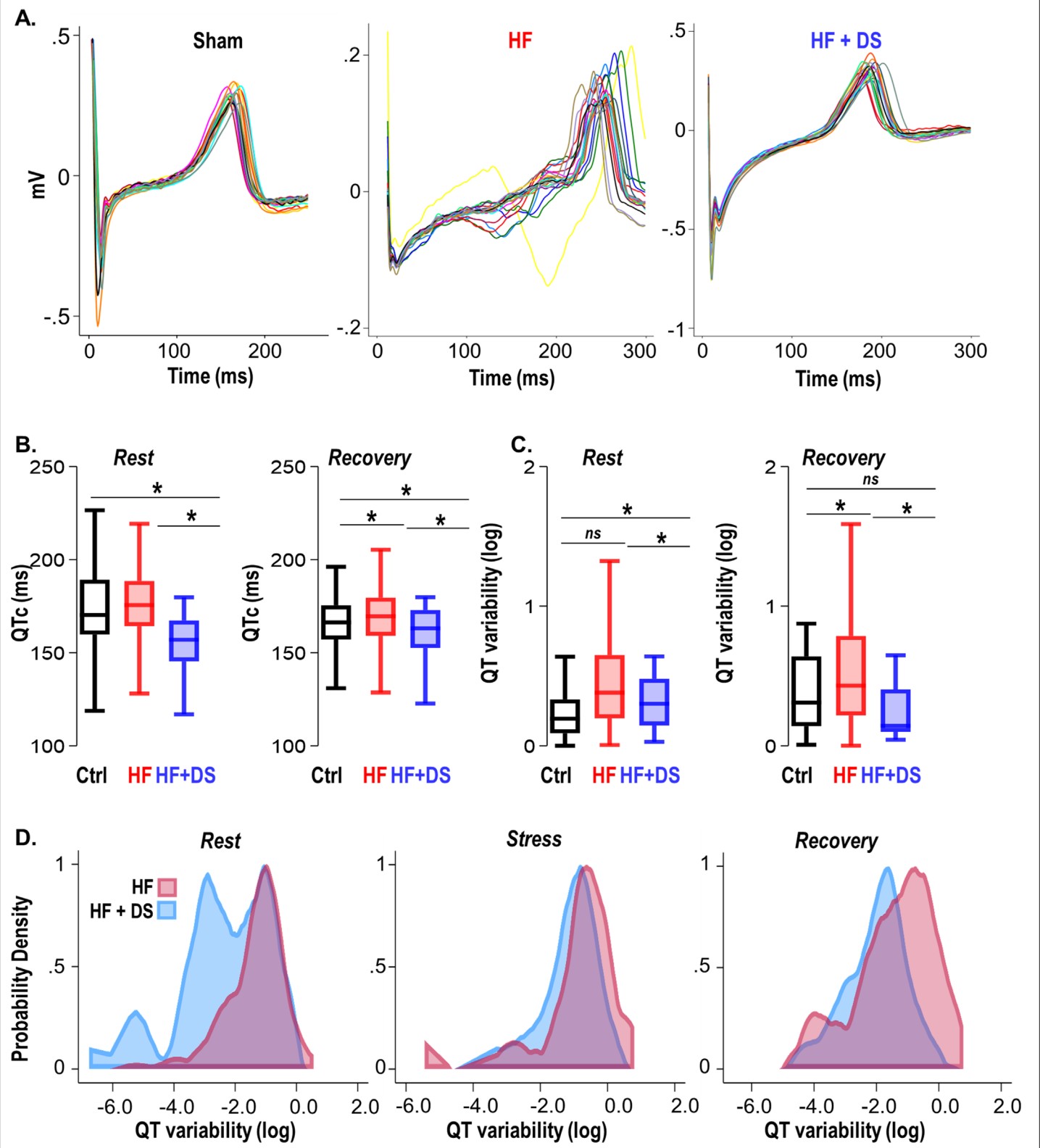

**Figure 5.** Dantrolene mitigates repolarization abnormalities, normalizes the QT variability, and reduces T-wave heterogeneity. (**A**) Representative QT segments recorded over 24 hr from Ctrl, heart failure (HF) and HF+DS show heterogeneity of T-wave repolarization in HF (red), but not in HF+DS (blue) animals. Dantrolene (DS) mitigates increased dispersion of repolarization in HF animals. (**B**) QTc prolongation in HF animals is significantly shortened by DS treatment (p<0.005), both at rest and post-stress recovery. The dispersion of repolarization was quantified by measurement of QT variability.

*Figure 5 continued on next page*

*Figure 5 continued*

(**C**) Increased QT variability predisposes the heart to ventricular tachycardia/fibrillation (VT/VF), which is prevented by DS therapy (p<0.0001). (**D**) Kernel density plots show that the QT variability index increases during rest, transient stress, and post-stress recovery. QT variability recovers quickly post-stress in the DS-treated group. A p<0.05 was considered significant; two-tailed log-rank analysis was performed on each group for each measure. (N: Control = 15, HF = 20, HF+DS = 8).

## Discussion

The novel findings of this study are: (1) Inhibition of RyR2 hyperactivity with DS abolishes VT/VF and SCD by normalizing repolarization abnormalities, QTc variability, and shortening prolonged QTc in non-ischemic HF; (2) dantrolene improves β-adrenergic signaling and chronotropic competency, thereby improving LV contractility; and (3) inhibition of RyR2 with DS not only prevents the progression of HF but also reverses chronic heart failure. These findings support the idea that the coupling between chronic sympathetic activity and oxidative stress makes RyR2 hyperactive. ROS increases the channel activity by decreasing the threshold for store overload-induced $Ca^{2+}$ release (*Jiang et al., 2004*; *MacLennan and Chen, 2009*; *Waddell et al., 2016*). This causes dispersion of calcium transients, reduces repolarization lability leading to spontaneous ectopic beats and VT/VF/SCD in HF (*Pogwizd et al., 1998*; *Szabo et al., 1995*).

Oxidative stress has been centrally implicated in HF. We and others have shown that the ROS dynamics affect the functioning of RyR2 (*Waddell et al., 2016*; *Gonzalez et al., 2010*; *Huang, 2021*; *Hamilton et al., 2020*; *Nikolaienko et al., 2018*). Increased energy demand and workload due to impaired contractile function and downregulation of antioxidant enzymes lead to high ROS accumulation (*Foster et al., 2016*). This is followed by an increase in use and shortage of reducing equivalents like NADPH/NADH, that further weakens the antioxidant defense (*Dey et al., 2018*; *Foster et al., 2016*). Our previous studies have confirmed that targeting mROS in failing hearts prevents VT/VF and SCD, and reverses heart failure. We see that mROS brings about several changes in protein function including oxidation (*Huang, 2021*), phosphorylation (*Wehrens et al., 2006*; *Dobrev and Wehrens, 2014*; *Shan et al., 2010*; *Uchinoumi et al., 2010*; *van Oort et al., 2010*), S-nitrosylation (*Gonzalez et al., 2010*; *Nikolaienko et al., 2018*; *Gonzalez et al., 2007*), and remodeling of the contractile machinery. Notably, the RyRs are in proximity to and tightly coupled with the mitochondrial network. They are readily modified by ROS (*Bertero and Maack, 2018*; *Kourie, 1998*; *Zima and Blatter, 2006*; *Terentyev et al., 2008*), via direct oxidation of thiol groups of cysteine residues (*Kourie, 1998*; *Zima and Blatter, 2006*; *Terentyev et al., 2008*) as well as ROS-mediated calcium-independent activation of CaMKII (*Curran et al., 2014*), increasing SR calcium leak (*Ellison et al., 2007*; *Curran et al., 2007*; *Morimoto et al., 2009*; *Ogrodnik and Niggli, 2010*) to promote VT (*Schlotthauer and Bers, 2000*), and increasing the channel activity by decreasing the threshold for store overload-induced $Ca^{2+}$ release (*Jiang et al., 2004*; *MacLennan and Chen, 2009*; *Waddell et al., 2016*; *Figure 7*).

In normal physiology, transient stimulation of the β-adrenergic receptors prompts a positive inotropic response to meet the physiological demands. This includes the opening of L-type calcium channels (LTCCs) and the release of cytosolic calcium from the SR, triggering contraction through the activation of ryanodine receptors. Tight regulation of $Ca^{2+}$ release from and reuptake into the sarcoplasmic reticulum is required for proper excitation-contraction coupling. However, in pathology, chronic activation of β-adrenergic receptors is maladaptive, increases ROS which remodels ion channels, modifies proteins, and worsens $Ca^{2+}$ leak from ryanodine receptors; these factors result in EADs or DADs that depose the heart to VT/VF, SCD, or injure myocardium; they may also impede contractile machinery. During diastole, the RyR2 still releases a small volume of $Ca^{2+}$ known as the leak. When this leak is continuous, like in failing hearts, the SR $Ca^{2+}$ is depleted and diastolic $Ca^{2+}$ is elevated, which weakens contractile force. At the same time, SR's capacity to retain the $Ca^{2+}$ is compromised. With the efflux of Na via the NCX, the exchanger imports $Ca^{2+}$, but this influx does not replenish SR $Ca^{2+}$ load, but rather contributes to the diastolic $Ca^{2+}$ .*Milberg et al., 2012*; *Remme and Bezzina, 2007*; *Valdivia et al., 2005*; *Takano et al., 2010* The increased Na load also depolarizes membrane potential and reduces conduction velocity, thus increasing anisotropy (*Bridge et al., 1990*; *Wier et al., 1994*). Additionally, intracellular [$Na^+$] accumulation at higher heart rates decreases $Ca^{2+}$ extrusion by NCX, which may now function in its reverse mode (*Smith, 1988*; *Langer, 1972*).

In the GP model of non-ischemic HF and SCD, we found that not only SERCA and RyR2 protein expression was downregulated, but the proteins were also hyperphosphorylated. Treatment with an

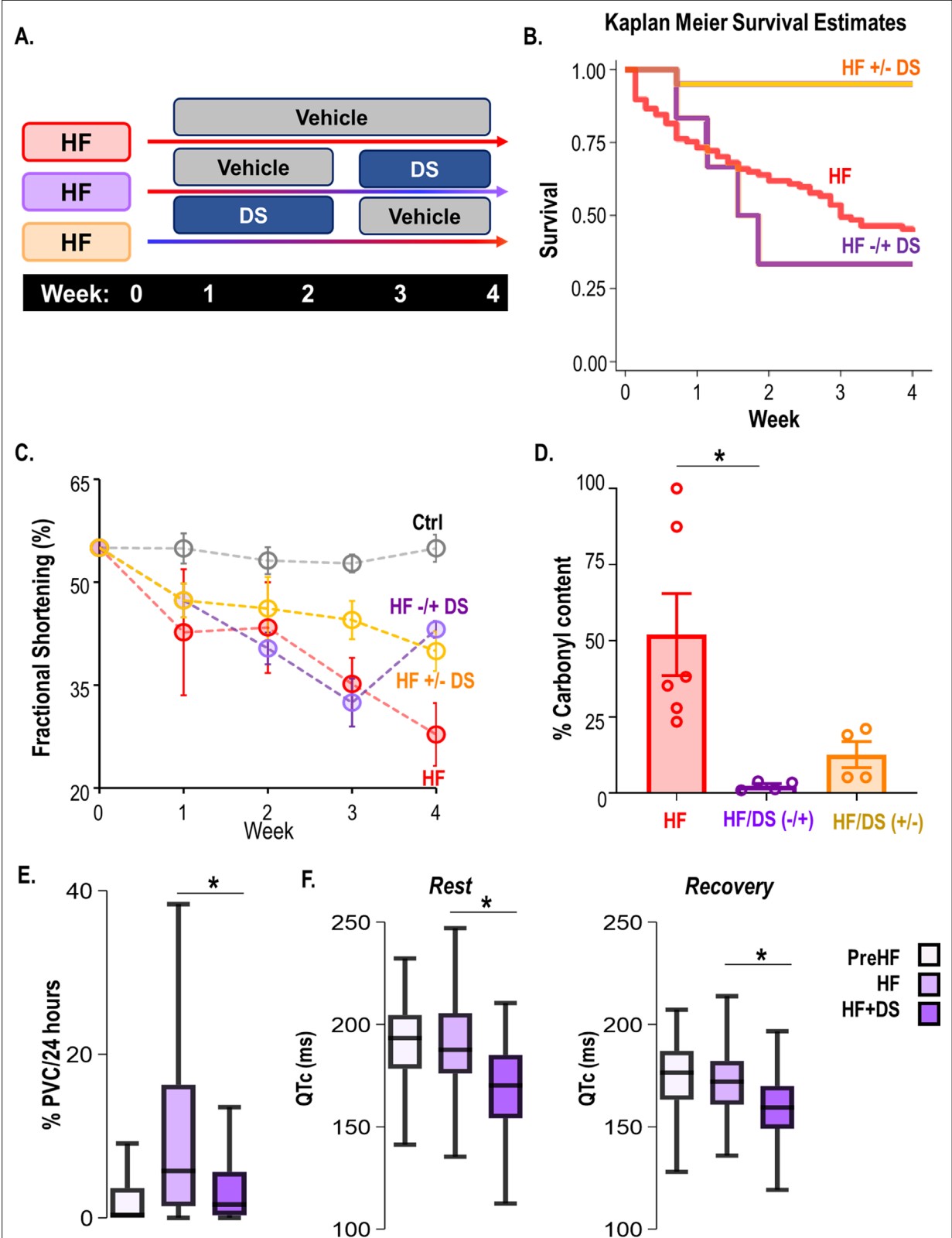

**Figure 6.** Dantrolene after heart failure (HF) development can mitigate ventricular tachycardia/fibrillation (VT/VF) and prevent sudden cardiac death (SCD). (**A**) Schematic illustrates a randomized 2 x 2 cross-over experimental design to examine the impact of dantrolene (DS) therapy. The protocols were as follows (**i**) only vehicle administered till the endpoint i.e., week 4 (HF), (ii) DS administered from the onset of HF until chronic contractile dysfunction is evident in the untreated control HF group (usually 3 weeks post-banding; HF +/−DS), or (iii) DS administered after HF had already

*Figure 6 continued on next page*

*Figure 6 continued*

developed in the HF group (3 weeks post-banding; HF−/+DS group). Echocardiography and ECG recordings were taken at week 2/3/4/5. (**B**) Kaplan Meier's survival curve showed that 50% of the HF animals experience SCD. Treatment with DS after HF development (around week 3) prevented additional SCDs and improved survival after starting DS therapy in the HFDS−/+group (purple). Discontinuation of DS, however, did not increase mortality in HF +/−DS group(yellow). (**C**) Serial echocardiography plot shows that DS reversed heart failure in HF−/+DS (purple) animals. The fractional shortening (FS)% in this group initially declined at a pace similar to HF (red) animals, but it quickly normalized as soon as DS therapy was started. However, in the HF +/−DS (yellow) group when the therapy was stopped, the FS% started showing a gradual decline, most likely due to the washout effect without additional SCD. (**D**) DS treatment pre or post-HF development alters ryanodine receptor 2 (RyR2) oxidation of HF animals (p<0.05). Treatment with DS reduced RyR2 oxidation even when therapy was initiated or discontinued after HF development. (**E**) Close examination of the HF−/+DS (purple) group, pre and post-DS therapy shows an increase in PVC load with the progression of HF. (**G**) The box plots summarize QTc pre-HF (light purple), during chronic HF (medium purple) and after DS therapy was initiated (dark purple). The HF−/+DS group displays all symptoms of HF including prolonged QTc after the development of HF (around week 3, medium purple box). After the start of therapy (dark purple box), QTc shortened at rest and recovery. A p<0.05 was considered significant; two-tailed log-rank analysis was performed on each group for each measure.

The online version of this article includes the following figure supplement(s) for figure 6:

**Figure supplement 1.** Dantrolene after heart failure (HF) development alleviates HF symptoms.

mROS scavenger prevented these alterations from occurring (*Dey et al., 2018*). Because ROS is a secondary messenger, a systemic antioxidant therapy may suppress cellular signaling. A target like RyR2, which is downstream of ROS and directly involved in $Ca^{2+}$ handling, could be of potential clinical interest. Post-translational modifications by oxidation of the RyR2 during the progression of HF causes leaky RyR2, which decreases SR $Ca^{2+}$ stores and elevated levels of diastolic calcium. Decreased SR $Ca^{2+}$ stores, leads to decreased $Ca^{2+}$ transient amplitude and duration during systole.

Our results show that chronic treatment with dantrolene in HF/SCD animal prevents lethal VT/VF, prevents HF, and delays the onset of cardiac decompensation in pressure overload. The effects of dantrolene on RyR2 was reflected by a decrease in oxidation of the RyR2 at the molecular level, a decrease in PVC burden, and an improvement in survival at the functional level in vivo. Pressure overload-induced structural alterations like increased fibrosis (*Sedej, 2014*). and progressive contractile dysfunction, thinning of ventricular walls and ventricular dilation were successfully ameliorated by chronic dantrolene treatment. Our previous work confirmed that sham controls (sham HF + isoproterenol challenge in the absence of pressure overload) did not significantly increase fibrosis, although there was a trend towards more (*Liu et al., 2014*; *Liu et al., 2021*). Although no mechanistic links between SR calcium leak and fibrosis has been explained, an association between reduced SR $Ca^{2+}$ and fibrosis has been previously reported (*Huang, 2021*; *Sedej, 2014*; *Nofi et al., 2020*). When dantrolene treatment was started after chronic HF was established, the contractile function and electrical properties of the heart started to recover and reverse.

Comprehensive analysis of the electrophysiological properties of the HF/SCD group shows that with chronic HF decrease in heart rate variability, QT prolongation, and increase in dispersion of repolarization. An increase in QT variability is due to the reduction of repolarization lability. Reduction of repolarization lability is associated with increased susceptibility of arrhythmogenesis in patients with prolonged QT and/or structural heart disease. With a high cytoplasmic $Ca^{2+}$ load, the inward sarcolemmal NCX current might significantly reduce the function of repolarization lability in calcium overloads and induce early (EADs) and delayed after-depolarization (DADs). Subsequently, the vulnerability to arrhythmias is increased. These local $Ca^{2+}$ handling abnormalities are reflected in beat-to-beat changes in ECG (*Shusterman et al., 2006*) T wave morphology and lability causing augmented QT variability (*Rosenbaum et al., 1994*; *Gold et al., 2008*). Dantrolene prevents SR $Ca^{2+}$ leak from RyR2 and normalize $Ca^{2+}$handling, suggesting that SR $Ca^{2+}$ leak is central to $Ca^{2+}$ induced triggered activity. Chronic dantrolene therapy enhanced chronotropic competence in response to increased metabolic demand in addition to correcting ventricular electrophysiological abnormalities, even when DS therapy started after HF had already been established.

Because the electrophysiological, $Ca^{2+}$ handling, autonomic, metabolic, and immune systems of the guinea pig are similar to that of humans, the findings in this study may be more relevant to humans as compared to other small animal models of SCD. For example, the ventricular action potential has a rapid upstroke and long plateau, defined by the balance of inward L-type $Ca^{2+}$ current (Cacna1c) and repolarizing currents carried by the inward rectifier K channel (Kcnj2), and the rapid (Kcnh2) and slow (Kcnq1/Kcne1) components of the delayed rectifier K current, as well as the long cellular action

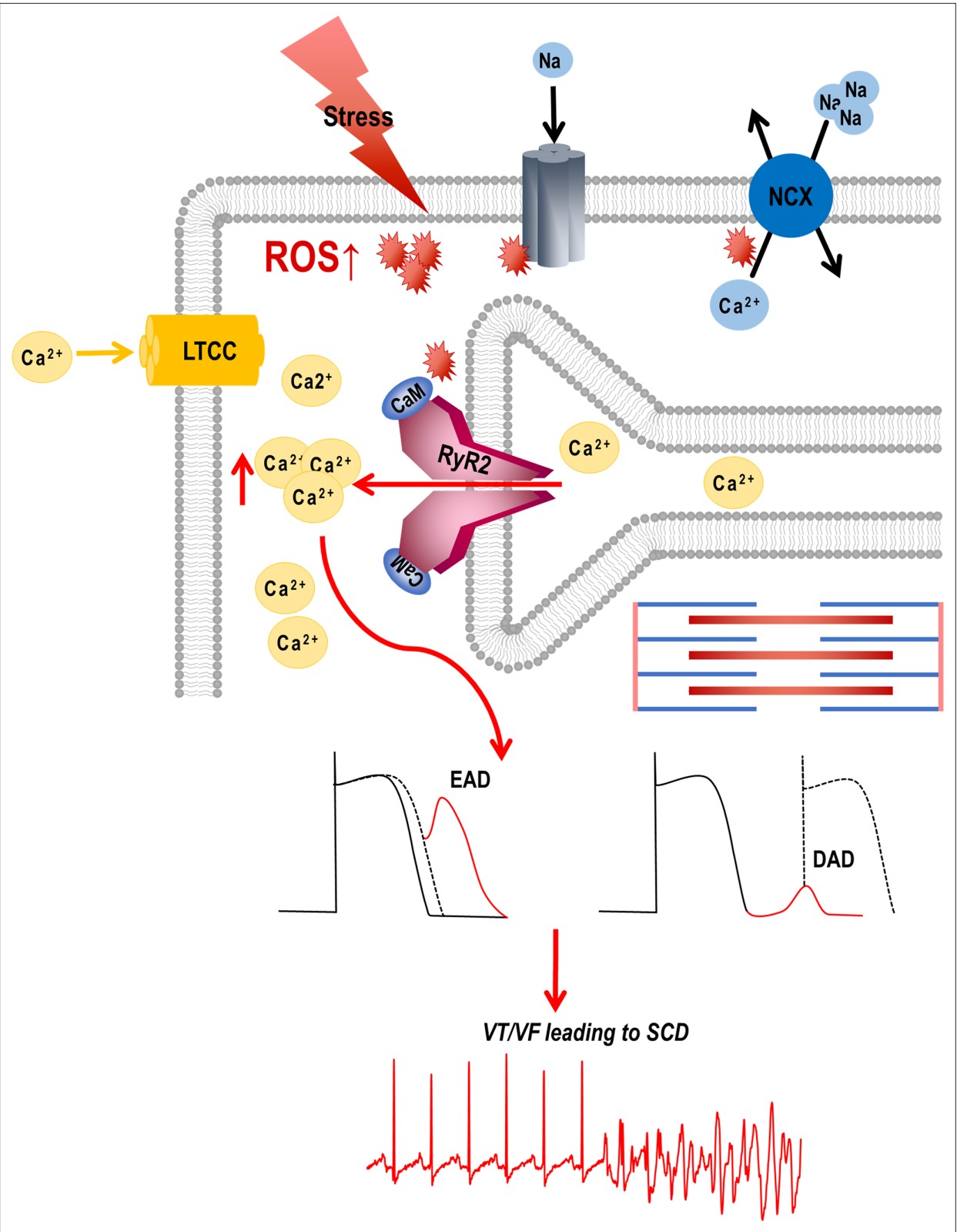

**Figure 7.** Graphical abstract. Stress-induced reactive oxygen species (ROS) increases Ca²⁺ overload via leaky ryanodine receptor 2 (RyR2) channels causing ventricular tachycardia/fibrillation (VT/VF) and SCD. External or internal stress releases substantial amounts of ROS in the cardiovascular system. This ROS can alter the oxidation/phosphorylation profiles of several electrocardiogram (EC) coupling proteins including RyR2. The altered RyR2 is leaky causing spontaneous calcium release into the cytosol causing early afterdepolarizations (EADs) and diastolic calcium overload causing delayed

*Figure 7 continued on next page*

*Figure 7 continued*

afterdepolarizations (DADs). This can ultimately lead to triggered activity and cause ventricular tachyarrhythmias (VT/VF) leading to sudden cardiac death (SCD).

potential plateau morphology that contributes to the prolonged QT interval of human heart failure. Excitation-contraction (EC) coupling displays a 60:35 balance between SR $Ca^{2+}$ reuptake and Na/$Ca^{2+}$ exchange for $Ca^{2+}$ removal on each beat, as well as a 3:1 β to α myosin heavy chain isoform ratio (*Miyata et al., 2000*; *Griffiths, 1999*). These properties are remarkably similar to the adult human heart and essential for studying how cardiac remodeling contributes to contractile dysfunction and VT/VF as HF develops and progresses.

## Conclusion and future directions

Damage caused by oxidative stress alters the proteomic profile of the cell, including altering expression levels as well as the structure and function of EC-coupling proteins. This study demonstrates that inhibition of $Ca^{2+}$ leak via RyR2 pauses the transition to heart failure by reducing cardiac fibrosis, increasing repolarization lability, and decreasing QT heterogeneity, improving chronotropic competency, ultimately improving LV contractility, and mitigating lethal VT/VF. Given that dantrolene is available for clinical use, future studies should evaluate its efficacy in humans with heart failure.

## Methods

### Animal model

All animal work followed IACUC-approved protocols at the respective institutions. A pressure overload model of HF and SCD was surgically generated with ascending aortic constriction (AC) and a daily bolus of low-dose isoproterenol (2 mg/kg/day) for β-adrenergic challenge. The surgical procedure and animal model have been previously described in detail (*Dey et al., 2018*). Briefly, animals were anesthetized with isoflurane then intubated. Ascending AC was accomplished by banding the aorta with a 18 mm diameter ligation clip. Throughout the anesthetic event, animals were continuously monitored for vital signs including ECG, pulse oximetry, and temperature. Sham animals received a thoracotomy without aortic banding. Daily isoproterenol bolus was delivered via a programmable iPRECIO pump (Primetech Corp., Tokyo, Japan) implanted in the peritoneal cavity. Animals were randomized to either vehicle or DS. The DS (25 mg/Kg, Patterson Vet catalog# 07-893-7963) was administered via the drinking water. All animals received an ECG transmitter implant (Data Sciences International) in the abdominal cavity and the leads were secured in a lead II axis arrangement. All devices and surgical equipment were gas sterilized (ethylene oxide) or autoclaved. A small subgroup of animals were subjected to a randomized crossover study. The following treatment groups were studied:

1. Normal Control
2. HF (AC + once daily β-adrenergic challenge till end point)
3. HF+DS (HF animals received DS therapy till endpoint)
4. HF+DS (+/-) (HF animals received DS therapy till week 3; DS was stopped after week 3)
5. HF+DS (-/+) (HF animals without DS therapy till week 3; DS therapy was started during week 3 and continued till endpoint)

Personnel involved in data collection and analysis were blinded to the treatment and non-treatment groups. All groups received similar incisions and thus, could not be distinguished based on these interventions. Each animal was assigned a unique computer-generated numeric ID. Some data were collected in part at Johns Hopkins University and the University of Cincinnati. Only male animals were used for this study because of heterogeneity in the phenotype of female animals. The progression of HF in female guinea pigs were delayed and the incidence of VT/VF/SCD was distinct from males. Females and ovariectomized females will be the focus of a follow-up study.

### Echocardiography

Echocardiography was performed on conscious animals using a L16-4HE Linear Array transducer (5.4 MHz-13.5 MHz) with a Mindray M9 Vet Ultrasound System in a blinded manner. Long-axis views were used to obtain two-dimensionally directed M-mode images. Echocardiography measurements

from M-mode images were used to determine fractional shortening, wall thickness, cardiac function, volumes, and ejection fraction of the different treatment groups. Three consecutive cardiac cycles by the leading edge-to-leading edge method were used to obtain the measurements. LV end-diastolic and end- systolic dimensions, and LV end-diastolic posterior wall thickness, were measured from the M-mode images, and left ventricular fractional shortening was calculated by the software.

### Electrocardiogram (ECG) analysis

Continuous serial ECG recordings were collected using the Data Sciences International, St. Paul MN, telemetry system. Ponemah software was used for data acquisition. Data was exported to and analyzed in a custom-developed MATLAB software (MathWorks, Inc, Natick, MA). ECG tracings were reviewed to quantify PVC, VT/VF burden and to adjudicate if death occurred due to SCD. ECG features were extracted from the recordings using the custom MATLAB algorithms and parameters were compared between the various treatment groups.

### RyR2 immunoprecipitation and oxidation assay from guinea pig left ventricle tissue

Snap frozen guinea pig LV tissue was lysed using RIPA lysis buffer (Sigma, cat# R0278), supplemented with phosphatase (Roche, cat# 4906837001), and protease inhibitors (Roche, cat# 4693124001). RyR2 was immunoprecipitated from tissue lysate using anti-RyR2 antibody (Invitrogen Cat# MA3916, 5 µL) in 0.5 mL RIPA buffer overnight at 4 °C as previously described *Hamilton et al., 2020*. Samples were incubated with Protein A/G Plus agarose beads (Santa Cruz, cat# sc-2003) for 1 hr at 4 °C and washed three times with RIPA buffer. To determine the oxidation status of RyR2, the Carbonyl Content Assay Kit (Abcam, cat #ab126287) was used, whereby carbonyl groups of immunoprecipitated RyR2 were derivatized to 2,4 dinitrophenylhydrazone (DNP) by reaction with 2,4 dinitrophenylhydrazine which can be quantified using a spectrophotometer at 375 nm.

### Masson's Trichrome (MT) staining

Connective and fibrotic tissues are stained blue, nuclei are stained deep brown to black, and the cytoplasm is stained red with MT staining. LV tissue was collected from HF animals with and without DS treatment and fixed in 10% neutral buffered formalin. The samples were sectioned and stained at the Translational Pathology Shared Resource at Vanderbilt University.

### Statistical analysis

All animal groups were age and weight-matched male and randomized to study groups. We use Stata 17 (Stata Corp LP, College Station, TX) for statistical analyses. The Kolmogorov-Smirnov test was used to evaluate the normality of distribution. For normal distributions, continuous variables were compared for two groups using unpaired t-test and ≥3 groups using ANOVA. Non-parametric tests were used for data not normally distributed. Kaplan Meier analysis was used to compare survival over time between groups. Bonferroni's correction was applied when a sample had multiple time points/ treatment measurements. A p-value of 0.05 was considered significant and is denoted with an asterisk (*) in the figures.

## Acknowledgements

This work was supported by American Heart Association Grants AHA19TPA34850151(to SD) and AHA19SFRN34830019 (to BCK), NHLBI/National Institutes of Health Grants NIH 4R00HK130662 (to DD), NIH DP2HL157941 (to DD), and Department of Defense Grants DOD-W81XWH-19-1-0640 (to SD), DOD-W81XWH-20-1-0701 (to SD and DD).

# Additional information

## Funding

| Funder | Grant reference number | Author |
|---|---|---|
| Congressionally Directed Medical Research Programs | W81XWH-20-1-0701 | Swati Dey |
| Congressionally Directed Medical Research Programs | W81XWH-19-1-0640 | Swati Dey |
| American Heart Association | AHA19TPA34850151 | Swati Dey |
| American Heart Association | 19SFRN34830019 | Bjorn C Knollmann |
| National Institutes of Health | 4R00HK130662 | Deeptankar DeMazumder |
| National Institutes of Health | DP2HL157941 | Deeptankar DeMazumder |

The funders had no role in study design, data collection and interpretation, or the decision to submit the work for publication.

## Author contributions

Pooja Joshi, Data curation, Formal analysis, Methodology, Writing – original draft, Writing – review and editing; Shanea Estes, Data curation, Formal analysis, Methodology, Writing – review and editing, Animal model development; Deeptankar DeMazumder, Resources, Formal analysis, Supervision, Funding acquisition, Investigation, Writing – original draft, Project administration, Writing – review and editing; Bjorn C Knollmann, Conceptualization, Supervision, Funding acquisition, Validation, Project administration, Writing – review and editing; Swati Dey, Conceptualization, Resources, Software, Supervision, Funding acquisition, Investigation, Writing – original draft, Project administration, Writing – review and editing

## Author ORCIDs

Pooja Joshi (ID) http://orcid.org/0000-0002-1415-1251
Swati Dey (ID) http://orcid.org/0000-0003-4692-6848

## Ethics

This study was performed in strict accordance with the Guide for the Care and Use of Laboratory Animals of the National Institutes of Health. All animals were handled according to IACUC-approved protocols (GP19M37, 19103101) at the respective institutions. All surgeries were performed under sodium pentobarbital anesthesia. Every effort was made to minimize any suffering or discomfort.

Reviewer #1 (Public Review): https://doi.org/10.7554/eLife.88638.3.sa1
Reviewer #2 (Public Review): https://doi.org/10.7554/eLife.88638.3.sa2
Author Response https://doi.org/10.7554/eLife.88638.3.sa3

# Additional files

## Supplementary files

• MDAR checklist

## Data availability

In accordance with institutional guidelines to ensure access to qualified researchers and applicable laws, regulations and policies governing data sharing and IP, applicable data be made available to collaborating researchers and submitted for peer-reviewed publication as per NIH standards. Genetically modified or mutant organisms were not generated in this project. Instead, non-genetic animal

models were generated with varying degrees of risk for spontaneous sudden death that mimic their human counterparts as follows: (a) thickened hypertensive hearts; (b) non-ischemic, dilated heart failure; and (c) controlled surgical and/or pharmacological interventions. Generated resources include animal tissue, proteome/phosphoproteome data and physiological data. For example, Western blot, RT-PCR, immunofluorescence, histology, electrocardiogram and echocardiography data will be available on request. Echocardiography data will be exported and shared as video and snapshot files in the .AVI and .TIFF formats, respectively. Algorithms and software will be made available in accordance with applicable NIH, IP and institutional policies on data and resource sharing. Sharing of unique research resources, including the sharing of biomedical research resources, will be governed by the performing institution's IP Office. The PI will be the point of contact for data sharing and management.

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
