## [Editor Report · eLife assessment]

This **important** study examined the use of dantrolene, a Ryanodine Receptor stabilizer, in slowing pathological progression of pressure-overload heart failure in a guinea pig model and reducing arrhythmias. **Convincing** data were collected and analyzed using validated methodology and can be used as a starting point for future studies of dantrolene in Ca2+ handling in ROS production and further deterioration of cardiac function in chronic heart failure.

---

## [Referee Report · Reviewer #1 (Public Review)]

The current study tests the hypothesis that inhibition of ryanodine receptor 2 (RyR2) in failing arrhythmogenic hearts reduces sarcoplasmic Ca leak, ventricular arrhythmias and improves contractile function. A guinea pig model of nonischemic heart failure (HF) was used and randomized to receive dantrolene (DS) or placebo in early or chronic HF. The authors show that DS treatment prevented ventricular arrhythmias and sudden cardiac death by decreasing dispersion of repolarization. The authors conclude that inhibition of RyR2 hyperactivity with DS mitigates the vicious cycle of sarcoplasmic Ca leak-induced increases in diastolic Ca and reactive oxygen species-mediated RyR2 oxidation. Moreover, the consequent increase in sarcoplasmic Ca2+ load improves contractile function.

In general, the study is well designed and the findings are likely to be of interest to the field.

---

## [Referee Report · Reviewer #2 (Public Review)]

Joshi et al. investigated the use of dantrolene, an RyR stabilizing drug, in improving contractile function and slowing pathological progression of pressure-overload heart failure. In a guinea pig model, they found that dantrolene treatment reduced cytosolic Ca2+ levels, improved contractility, reduced the incidence of arrhythmias, reduced fibrosis, and slowed the progression of heart failure. Importantly, delaying treatment until 3 weeks after aortic banding (when heart failure was already established) also resulted in improvements in function and decreased arrhythmogenesis. While some of the mechanistic details remain to be worked out, the data suggest that improving intracellular Ca2+ handling can break the vicious cycle of sympathetic activation, ROS production, and further deterioration of cardiac function.

The functional ECG and echo data are convincing, and very clearly demonstrate the positive effects of dantrolene in heart failure. This is important because dantrolene is already FDA-approved to treat malignant hyperthermia and muscle spasms, so repurposing this drug as a heart failure therapeutic might have a straightforward path to clinical implementation. This also highlights the non-specific nature of dantrolene to interact with RyR1, and therefore, potential side effects. However, this does not detract from the main proof-of-concept demonstrated here.

The guinea pig model employed here is also a strength, as the guinea pig has intracellular Ca2+ handling and ionic currents that are much more similar to human (vs. a murine model, for example).

One weakness is the exclusion of female animals from the study. The authors report more heterogeneity in the progression of HF in the female guinea pig model, however it will be very important to determine effects of dantrolene in the female heart, as there are considerable known sex differences in intracellular Ca2+ handling and contractility. Therefore, it is possible that dantrolene could have sex-dependent effects.

---

## [Author Response]

The following is the authors’ response to the original reviews.

We thank the reviewers for their positive feedback and very helpful comments. We agree that this manuscript focuses primarily on functional outcomes and phenotypes. The studies were designed to address an important clinical question, i.e., repurposing dantrolene for the treatment of ventricular tachyarrhythmias and the prevention of sudden cardiac arrest. Thus, the current manuscript emphasizes in vivo studies over in vitro studies.

However, we also acknowledge the need for additional mechanistic studies. We are in the final stages of submitting a second manuscript in which we dissect the underlying mechanisms through detailed in vitro studies of mitochondrial antioxidant capacity, reactive oxygen species, phosphorylation of ryanodine receptors, autonomic dysfunction, beta-adrenergic signaling, etc. that are beyond the scope of the current manuscript.

Additionally, a third manuscript in progress focuses on the mechanistic link between ion channels, dispersion of repolarization, and sudden cardiac death. We previously reported the preliminary results in abstract form (Circulation Research. 2019;125:A102). Briefly, current-voltage relationships from patch clamp studies of isolated LV myocytes revealed that pressure-overload stress strongly reduced K currents, including IK1, IKs, and IKr. These changes were driven by downregulation of K channels and their components at the mRNA level. As expected, the reduced K currents destabilized the resting membrane potential, especially in phases II and II of the cardiac action potential, and reduced repolarization reserve. Scavenging mitochondrial ROS stabilized repolarization, suggesting mROS is the upstream driver of K channel downregulation. However, we have not specifically tested whether dantrolene stabilizes repolarization via the same mechanism. As such, we agree that "lability" or "dispersion" are more precise terms than "reserve" for the phenomenon reported in the present manuscript, and we have made these changes. Thank you for pointing this out. We have also changed the title accordingly.

The present study investigates the effect of dantrolene on male animals. We agree that we need to evaluate the effect on females, especially because females have historically been underrepresented in studies of sudden cardiac arrest. Based on our preliminary studies, female animals exhibit increased variability in their phenotypic response to pressure-overloaded stress. Given the importance of this issue, we will examine the sex differences in carefully controlled future experiments, including the effect of dantrolene in females controlled for hormonal effects (e.g., with and without oophorectomy).